# Modeling non-uniform uncertainty in Reaction Prediction via Boosting and Dropout

## Abstract

Reaction prediction has been recognized as a critical task in synthetic chemistry, where the goal is to predict the outcome of a reaction based on the given reactants. With the widespread adoption of generative models, the Variational Autoencoder (VAE) framework has typically been employed to tackle challenges in reaction prediction, where the reactants are encoded as a condition for the decoder, which then generates the product. Despite effective, these conditional VAE (CVAE) models still fail to adequately account for the inherent uncertainty in reaction prediction, which primarily stems from the stochastic reaction process. The principal limitations are twofold. Firstly, in these CVAE models, the prior of generation is independent of the reactants, leading to a default wide and assumed uniform distribution variance of the generated product. Secondly, reactants with analogous molecular representations are presumed to undergo similar electronic transition processes, thereby producing similar products. This hinders the ability to model diverse reaction mechanisms effectively. Chemical reactions are inherently stochastic processes. Reactions may have multiple potential pathways leading to different products, and some pathways might be more probable than others, depending on the reactants and condition involved. Since the variance in outcomes is inherently non-uniform, we are thus motivated to develop a framework that generates reaction products with non-uniform uncertainty. Firstly, we eliminate the latent variable sampled from an independent prior in previous CVAE models to mitigate uncontrollable noise. Instead, we introduce randomness into product generation via *boosting* to ensemble diverse models and cover the range of potential outcomes, and through *dropout* to secure models with minor variations. Additionally, we design a ranking method to union the predictions from boosting and dropout, prioritizing the most plausible products. Experimental results on the largest reaction prediction benchmark USPTO-MIT show the superior performance of our proposed method in modeling the non-uniform uncertainty compared to baselines.

## 1 Introduction

Reaction outcome prediction is one of the fundamental problems in computer-aided organic synthesis. The aim of this task is to predict the most likely products formed given a set of reactants and reagents (Corey & Wipke, 1969; Coley et al., 2017). A number of works have aimed at inferring products from a reaction. These methods can be grouped into two main categorizes: template-based (Segler & Waller, 2017a;b) and template-free (Jin et al., 2017). Template-based methods are typically rule-based and rely heavily on expert knowledge for generalizing reaction patterns and mechanistic pathways. The intense knowledge requirement and the labor-intensive nature of template creation constrain these methods' scalability and adaptability, struggling with novel or unconventional reactions. To tackle this challenge, template-free methods are proposed to leverage deep generative models to predict products from reactants. Traditionally, these template-free methods formulate reaction prediction as graph-level sequential editing or Seq2Seq translation tasks with autoregressive decoding (Coley et al., 2019; Somnath et al., 2021), which poses two major limitations. 1) One failed step in this successive procedure may invalidate the entire prediction (Schwaller et al., 2019; Ucak et al., 2021; Tu & Coley, 2022); and 2) They lack the efficiency of parallel decoding. Hence, a non-autoregressive reaction prediction (NERF) method has been proposed recently (Bi et al., 2021). This method uses Graph Neural Networks and Transformers to predict the redistribution of electrons

between the reactants and products. During inference, given the reactants, this method can predict the electron transition matrix, which, when added to the reactant matrix, yields the product matrix. This method has achieved state-of-the-art performance while supporting parallel decoding.

Nevertheless, the conditional VAE (CVAE) framework employed by NERF fails to adequately account for the inherent **non-uniform uncertainty** in reaction prediction, especially concerning the variance in the distribution of the generated products. Given the intrinsic stochastic nature of chemical reactions, multiple potential pathways can exist for the same reactant, each leading to different products. The probability of these pathways can vary significantly as well, depending on the reactants and condition involved. Thus the variance in outcomes is inherently non-uniform. Figure 1 shows the examples of reaction with high and low uncertainty. It is evident that, in reactions charac-

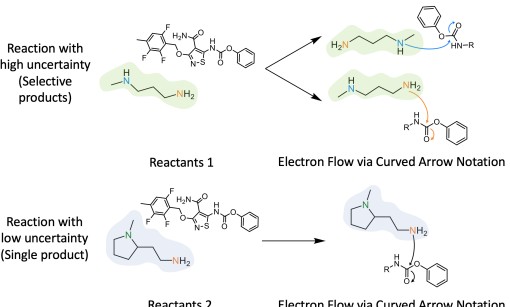

Figure 1: Non-uniform uncertainty in chemical reactions.

terized by high uncertainty, the two mechanisms of electron redistribution are substantially different. This creates a unique challenge to strike a balance between extreme and mild variances in outcomes; that is, to handle reactions with lower-level uncertainty effectively while concurrently predicting diverse outcomes accurately for a single reactant in scenarios with high-level uncertainty.

To illustrate this challenge, we present a two-dimensional plot featuring pairs of reactants alongside their corresponding electron redistributions to predict. As shown in Figure 2(a), for identical or similar reactants (sharing the same x-axis coordinate value), the electron redistributions may yield a singular outcome (indicative of low uncertainty) or multiple possibilities (highlighted by samples with varying y-axis coordinate values). A deterministic prediction can only handle a one-to-one mapping between reactants and electron redistribution (see Figure 2(b)). CVAE architectures incorporating an independent prior (unrelated to the reactants) in generation cannot manage non-uniform uncertainties effectively. This leads to incorrect or invalid predictions in cases with low-level uncertainty and a lack of varied predictions in scenarios characterized by high-level uncertainty, as illustrated in Figure 2(c).

The optimal mapping function, capable of managing the non-uniform uncertainty depicted in Figure 2(a), should exhibit the following key features: 1) in low uncertainty scenarios, it accurately maps reactants to the singular, predictable outcome; and 2) In high uncertainty scenarios, it predicts a plausible list of products for the given reactant, reflecting the range and diversity in possible electron redistributions. Drawing upon the efficacies of boosting and dropout during inference, as depicted in Figure 2(d-e), our approach integrates *Boosting Training* and *Dropout* to respectively manage large and small range uncertainties. Concretely, we first replace the CVAE architecture with a deterministic GNN+Transformer architecture to mitigate the uncontrollable noise that CVAE suffers from. This eliminates noise from independent prior in the decoder during inference, leading to more accurate predictions. Then we employ *Boosting Training* to decompose the complex reaction mapping to several independent mappings. This is complemented by the application of *Dropout*, managing the small-range uncertainty, as shown in Figure 2(f). The underlying principle here is that dropout enables the creation of diverse neural networks by allowing a random subset of neurons to be nullified. Each distinct neural network conceived in this manner represents a spectrum of potentialities for electron transition within a smaller range. By integrating *Boosting Training* and *Dropout*, we produce an array of model variations, enabling a comprehensive exploration of reaction spaces. A simple ranking method is designed to prioritize most plausible products predicted by the model variations, maintaining an optimal balance between accuracy and diversity in predictions.

We summarize our contributions as follows:

- To the best of our knowledge, we are the first to investigate the issue of non-uniform uncertainty in reaction prediction tasks. This is a distinct and specialized challenge presented to generative models when applying AI to scientific discovery.

- We innovatively resolve identified challenges by utilizing boosting and dropout to model varied uncertainties.

- Our method's efficacy is validated through experiments on the USPTO-MIT dataset (Jin et al., 2017), demonstrating consistent advancements over existing baselines.

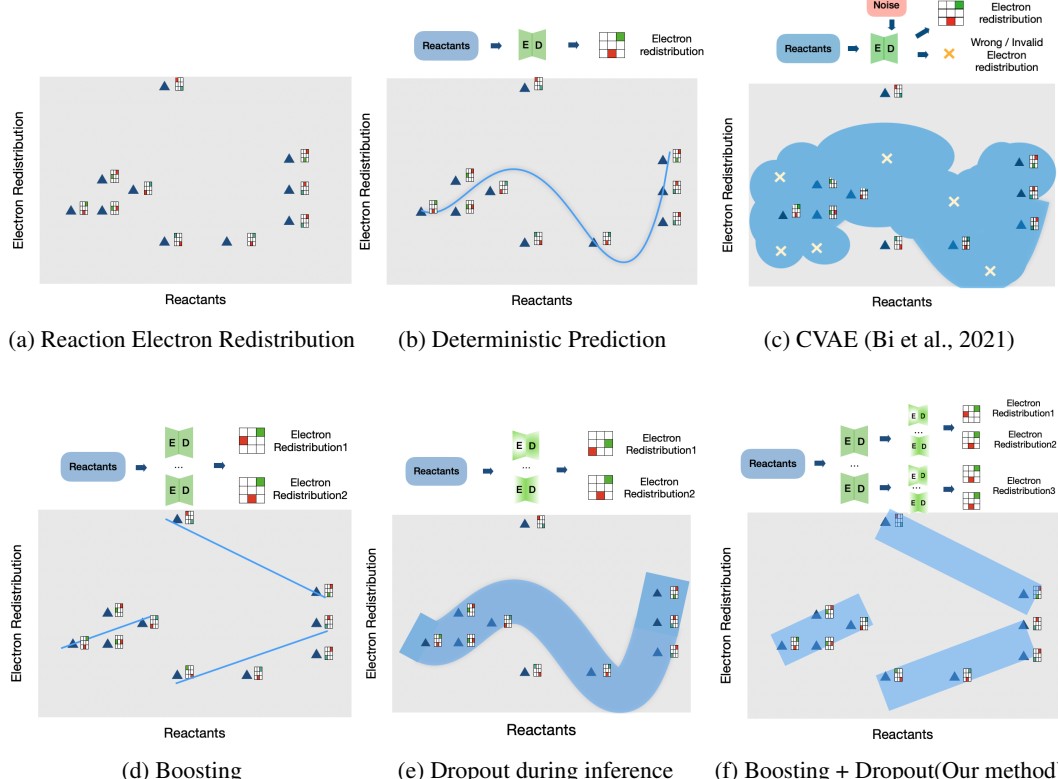

Figure 2: (a) Illustrates pairs of reactants with their corresponding electron redistributions to predict. The electron redistributions are illustrated by an atom-atom adjacency matrix; units situated at the $(i, j)$ position are marked in red to denote an electron deletion (bond-breaking), while those in green represent an electron insertion (bond-formation). (b-f) Showcase the mapping functions learned by various methods, with the respective model architectures depicted above ("E" denotes the Encoder and "D" denotes the Decoder. The partially lit model symbol demonstrates the application of dropout).

## 2 RELATED WORK

**Reaction Prediction:** Machine learning methods for reaction prediction can be categorized as template-based and template-free. Template-based methods (Segler & Waller, 2017a;b), although effective, rely heavily on expert knowledge and tend to be less generalizable. Among template-free methods, most solutions are autoregressive methods which either model reactions as a sequence of graph edits (Shi et al., 2020; Coley et al., 2017) or as a seq2seq translation (Liu et al., 2017; Schwaller et al., 2019; Guo et al., 2023). To address the drawbacks of autoregressive generation, the NERF model (Bi et al., 2021) is proposed to model electron redistribution in a non-autoregressive way. ReactionSink (Meng et al., 2023) extends NERF by imposing two essential rules of electron redistribution during generation. Both non-autoregressive methods are based on CVAE architecture and unable to handle the non-uniform uncertainty problem in the reaction prediction.

**Boosting:** In earlier years, boosting has been widely used in machine learning methods such as Adaboost (Schapire, 2013) or Gradient Boosting Decision Tree (Ke et al., 2017), which combine a set of weak models to a single stronger model. During the boosting phase, individual models are trained with varying sample weights, resulting in complementary and diverse models. Unlike prior applications focusing mainly on minimizing the bias error, our motivation is to leverage the model diversity during boosting phase to model the large-range uncertainty in the reaction prediction task.

**Dropout:** Dropout (Srivastava et al., 2014) is a common technique used to improve the generalization of the neural networks. It operates by randomly omitting a unique subset of neurons during each training iteration, preventing the network from becoming overly sensitive to the specific weights of neurons. In our work, we deviate from the conventional motivation behind Dropout. We utilize the

variability induced by selecting different neuron subsets through Dropout to model the small-range uncertainty present in the reaction prediction task. Thus, contrasting with conventional approaches that implement Dropout exclusively during training, we incorporate it during the inference phase. This approach facilitates the construction of slightly divergent models in each sampling process, allowing us to explore subtle differences in the predicted outcomes.

## 3 THE PROPOSED SOLUTION

### 3.1 NOTATIONS AND PROBLEM DEFINITION

Following the definition of previous reaction prediction methods (Bi et al., 2021), we formulate the reaction prediction as a transformation from reactants $G^r = (V^r, E^r)$ to products $G^p = (V^p, E^p)$. $V^r$ and $V^p$ denote the atoms, and $E^r$ and $E^p$ denote the number of shared electrons between atoms in the reactants and products, respectively. For example, $E_{ij}$ denotes the number of shared electrons (1 denotes the single bond and 2 denotes the double bond) between atom $i$ and atom $j$. The aim of the reaction prediction task is to learn a function $f$ that can predict the potential product list $P$ given the reactants $G^r$. The ground-truth product $G^p$ should be at the top of the predicted product list $P$.

### 3.2 CVAE WITHOUT INDEPENDENT PRIOR

As discussed earlier, the CVAE architecture falls short in accounting for the non-uniform uncertainty in reaction prediction. To address this, we first discard the prior sampling process and retain only the Encoder-Decoder part (see Figure 2b). The Encoder can be implemented by Graph Neural Networks (GNN) (Scarselli et al., 2008) to capture the local dependency between atoms in $G^r$, followed by a Transformer (Vaswani et al., 2017) to model the long-range dependency between atoms.

After encoding, the reactants $G^r$ are represented by vector $h^r$. Note that the latent variable sampled from an independent prior is omitted. Thus, only $h^r$ is fed to the Decoder for product prediction. Following the setting in NERF, PointerNet (Vinyals et al., 2015) is employed to compute the probabilities of an electron flow between atoms. Based on the Octet Rule, the number of active valence electrons in an atom is generally at most 8. Therefore, 16 independent PointerNets are used to compute the attention weights between atoms to represent the *BondFormation* $w_{ij}^{(+d)}$ and *BondBreaking* $w_{ij}^{(-d)}$, where $w_{ij}$ denotes the attention weights (i.e. probability of an electron flow from $i$ to $j$), and $d = 1, 2, \cdots, 8$. The values $w_{ij}^{(+d)}, w_{ij}^{(-d)}$ are all between 0 and 1.

Based on the *BondFormation* $w_{ij}^{(+d)}$ and *BondBreaking* $w_{ij}^{(-d)}$, the overall bond changes between reactants and products denoted by $\Delta w_{ij}$ can be calculated by:

$$\Delta w_{ij} = \sum_{d=1}^{8} w_{ij}^{(+d)} - \sum_{d=1}^{8} w_{ij}^{(-d)}. \tag{1}$$

Adding $\Delta w_{ij}$ to the reactant edges $E^r$, we can obtain the predicted product edges $\hat{E}^p$. Since the prior has been removed, the overall loss is defined only for product prediction (no KL-divergence):

$$L = \sum_{(i,j) \in E^p} \left( E_{ij}^p - \hat{E}_{ij}^p \right)^2. \tag{2}$$

### 3.3 BOOSTING

After removing the prior, CVAE becomes a deterministic model. It is thus necessary to introduce appropriate randomness for generation purpose. As shown in Figure 3, Boosting and Dropout techniques are integrated to manage the non-uniform uncertainty in generation. In this section, we will introduce the design of the boosting strategy in the training phase and inference phase.

#### 3.3.1 BOOSTING IN TRAINING PHASE

The key to designing the boosting strategy is to build a set of models $F = \{f_1, f_2, \ldots, f_n\}$ that are diverse and able to cover the large-range uncertainty.

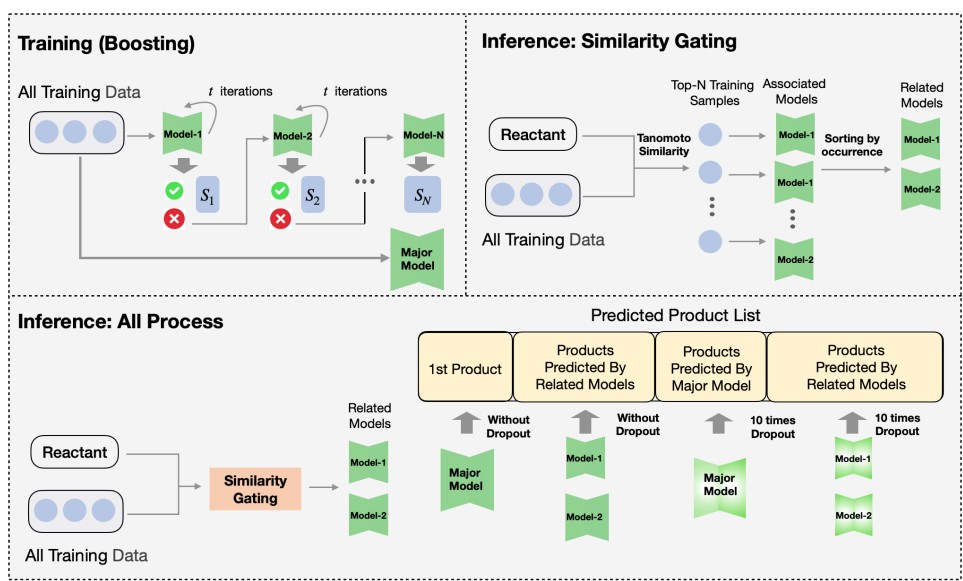

Figure 3: The overall framework. The application of dropout is illustrated by partial lighting.

**Warm-start.** Before launching the boosting training, we prepare the encoder-decoder network through a warm-up phase, in which the model is trained for several iterations utilizing the entire training dataset. This warm-start phase is crucial to ensure that the models, when engaged in boosting training, are proficient in recognizing and applying basic reaction prediction transformations. Consequently, this ensures that the predictive capabilities of all models $f_i$ are maintained at a satisfying level.

**Recording.** Inspired by Adaboost (Schapire, 2013), we train the model by different subsets of training data in each boosting iteration to obtain a set of diverse models. In traditional Adaboost iterations, the method discriminates between correctly classified and misclassified samples, elevating the weights of the misclassified and reducing those of the correctly classified for the next iteration. However, within our context of reaction prediction, the output from each model is interpreted as a predicted product, which is incompatible with the approach of adjusting sample weights for upcoming iterations prevalent in classification tasks. Therefore, we modify the soft-sampling by weights typically seen in traditional Adaboost and employ hard-sampling instead. Any samples predicted correctly are omitted from the sample pool in the following iteration. Consequently, we maintain a record of samples accurately predicted after each iteration,

$$Correct_i = \{(G^r, G^p) \in D_i \mid L(G^r, G^p) \leq \delta\} \tag{3}$$

where $D_i = \{(G_1^r, G_1^p), (G_2^r, G_2^p), \ldots, (G_n^r, G_n^p)\}$ is the training set in the $i_{th}$ iteration, $L(G^r, G^p)$ is the loss of the sample, and $\delta$ is for determining the prediction correctness (empirically set to 1e-3).

**Training Set Updating.** Given the unstable performance of the generative model, we assess the "correctness" of predictions on a single sample across $t$ iterations, instead of basing our judgment on a single iteration. A sample is only excluded from the training set if it is consistently correctly predicted over the $t$ iterations:

$$D_{i+1} = D_i - (Correct_i \cap Correct_{i-1} \cap \cdots \cap Correct_{i-t+1}) \tag{4}$$

where $D_{i+1}$ denotes the training set used for the next iteration, $D_i$ is the training set of current iteration (i.e., $i_{th}$ iteration). During updating, we also save the current model $f_i$ and add it to the final set of models $F$. Additionally, we store the correct training data $S = (Correct_i \cap Correct_{i-1} \cap \cdots \cap Correct_{i-t+1})$ for future usage.

**Termination.** Boosting training continues until it reaches the maximum iteration. The model list $F$ and the corresponding training set $S$ are then obtained. For each sample in $S$, we then identify an $f_i$ that can accurately process it. This association between samples and models is represented as $\tau$.

**Major Model Training.** The models within $F$ exhibit substantial diversity, each tailored to predict specific subsets of samples, leading to compromised generalization on unseen datasets. To counteract

this, we train a Major Model, $f_{major}$, by utilizing the entirety of the data until convergence, aiming to encapsulate all electron redistribution patterns. While it excels in generalization ability due to its comprehensive learning scope, $f_{major}$ tends to lack in diversity. Consequently, $f_{major}$ serves as a complementary entity to the boosting models in $F$, enhancing the overall generalization capabilities.

### 3.3.2 SIMILARITY GATING

In this section, we outline our methodology to route test reactants to the appropriate models in $F$ during inference. Unlike the original boosting method intended for classification or regression where outputs are numerical, our scenario necessitates product lists. Thus, our gating method ensures test reactants align with suitable models, prioritizing the ground truth product. Illustrated in Figure 3, we implement a similarity gating module during sampling, hypothesizing molecules with akin properties share analogous reaction mechanisms and, hence, fit similar prediction models.

For each test reactant $G^r$, Tanimoto Similarity (Bajusz et al., 2015) of the Morgan Fingerprint (Capecchi et al., 2020) (radius=2) computes the molecular scaffold similarity between $G^r$ and all training samples in $S$. Subsequently, the Top-N most analogous training molecules to the test reactants are selected, denoted as $TopN$. Reactants-to-models mapping obtained earlier $\tau$ then determines corresponding models $f_i$ for each reactant in $TopN$, returning them in descending order of occurrence for subsequent prediction phase.

### 3.4 DROPOUT DESIGN

The boosting stage generates a suite of diversified models capable of addressing extensive-range uncertainties; however, each $f_i$ remains deterministic. To manage finer, small-range uncertainties, we integrate the dropout technique. Initially deployed to regulate deep neural network training, prior dropout studies (Srivastava et al., 2014) mainly focus on its regularization and consistency during training for deterministic tasks. We propose a different approach, hypothesizing that dropout at each iteration yields subtly variant neural network weights and, consequently, differing attention weights between atoms, capturing fine-grained uncertainties in electron redistribution during reactions.

This contrasts previous works in reaction prediction, which model uncertainty explicitly through classifier probabilities or latent variables. We suggest utilizing dropout during inference to model the intricate uncertainties inherent in chemical reactions. We now delve into the specifics of our approach.

**What to drop.** We directly incorporate the dropout layer from the NERF model, applying dropout to the GNN encoder for molecules and the self-attention layer of the Transformer Encoder. This strategy ensures the most important neural network weights are subject to dropout.

**When to drop.** Different from the NERF model, which only uses dropout during training, we additionally apply dropout during inference. This modification implies that the outputs from the same model may vary with each inference iteration.

**How to drop.** Dropout is activated during inference. Given that the dropout layers are influenced by the random seed, we alter the random seed for each inference iteration to introduce variability in the predictions. This process repeats 10 times to generate the list of predicted products for each model.

### 3.5 RANKING IN THE FINAL PREDICTION PHASE

In the final-stage determination of the product for a given reactant, we utilize the models $f_i$ as selected by the similarity gating method (refer to Section 3.3.2), along with the Major Model $f_{major}$. Each produces predictions without dropout once and with dropout 10 times. The aggregate of these predictions is diverse, varying in reliability and accuracy. Different from the original boosting method which applies a weighted sum to the prediction of each model to get one single predicted value, our model generates a number of product candidates that need to be ranked to move the most likely product candidates to the front of the list. The ranking of product candidates is determined based on the positional order of the prediction models $f_i$. As explained in Section 3.3.2, the selected $f_i$ are ordered according to the count of training molecules in $TopN$ associated with them. For each time of dropout, the potential product list is updated by including the ranked candidates.

## 4 EXPERIMENTS

### 4.1 EXPERIMENT SETTINGS

**Datasets and preprocessing.** Same as most previous work (Coley et al., 2019; Schwaller et al., 2019), we evaluate our method on the current largest public reaction prediction dataset USPTO-MIT (Jin et al., 2017), which removes duplicates and erroneous reactions from the original data proposed by (Lowe, 2012). There are 479K reactions in this dataset. Follow the preprocessing of the NERF method (Bi et al., 2021), 0.3% of the USPTO-MIT reactions do not satisfy the non-autoregressive learning settings. Thus we also remove such reactions from both training and testing and subtract the predictive top-k accuracy by 0.3% as the final accuracy of our model.

**Implementation details.** We implement our model by Pytorch (Paszke et al., 2019) and conduct all the experiments on a Linux server with GPUs (4 Nvidia V100). As we mentioned in Section 3.2, we reuse the encoder and decoder architectures from the NERF model. To ensure a fair comparison to previous non-autoregressive methods, we follow the same detailed settings as the previous model. We set the number of self-attention layers in the Transformer encoder and decoder to 6, and the number of attention heads is 8. We also set the node embedding dimension to 256 and use the AdamW optimizer (Loshchilov & Hutter, 2017) at a learning rate $10^{-4}$ with linear warmup and learning rate decay. For the boosting training, we set the maximum boosting training iteration to 100 and the update interval $t$ to 2. For the similarity gating, we set the $N$ to 10 for selecting $TopN$. The implementation code will be made publicly available upon the acceptance of the paper.

**Evaluation metrics.** Following (Coley et al., 2019; Schwaller et al., 2019; Bi et al., 2021; Meng et al., 2023), we use the Top-K accuracy to evaluate the performance of our model and baselines. The Top-K accuracy measures the percentage of reactions that have the ground-truth product in the top-K predicted products. Following the previous setting, the $K$ is set as: $\{1, 2, 3, 5, 10\}$. For test reactants that have more than two products in the USPTO-MIT dataset, we assess prediction performance by calculating the reactant-wise HitRate, which is the percentage of products appearing in the predicted products that are also in the ground truth. It also means how many of the multiple ground truth products are correctly identified by the model among its predictions.

### 4.2 BASELINES

We compare our method with the following baselines which are classified into four categories:

**Template-based.** Symbolic (Qian et al., 2020) introduces symbolic inference which is based on chemical rules to the neural network to do reaction prediction.

**Two-Stage.** WLDN (Jin et al., 2017) firstly recognizes a group of reaction centers and corresponding bond configurations and then ranks them to obtain the final potential products.

**Autoregressive.** GTPN (Do et al., 2019) converts the reaction prediction problem as a sequence of graph transformations problem and then applies policy networks to learn such transformations; MT-based (Schwaller et al., 2019) treats the reaction prediction problem as a machine translation task from the reactants SMILES strings to products SMILES strings and applies transformer-based modeling; MEGAN (Sacha et al., 2021) models reaction prediction as a graph editing task and predicts the edit sequences autoregressively; MT (Schwaller et al., 2019) is based on MT-base and additionally apply data augmentation techniques on the SMILES; Chemformer (Irwin et al., 2022) is also a transformer-based model and it additionally pretrains the SMILES encoder with three self-supervised tasks; Sub-reaction (Fang et al., 2022) achieves the substructure-aware reaction prediction via modeling the motif of the molecule graphs; Graph2Smiles (Tu & Coley, 2022) combines the graph encoder and transformer encoder to achieve permutation-invariance encoding of molecule graphs in the reaction prediction; AT × 100 (Tetko et al., 2020) is also based on transformer and applies more data augmentations on SMILES.

**Non-autoregressive.** NERF (Bi et al., 2021) is the first to model reaction prediction in a non-autoregressive way; ReactionSink (Meng et al., 2023) is based on NERF and integrates two essential chemical rules of electron redistribution to NERF via Sinkhorn's algorithm. Both of them are based on the CVAE architecture and leverage the latent variable of a prior to introduce randomness, leaving the non-uniform uncertainty of the reaction prediction out of consideration.

Table 1: The Top-K Accuracy % on the USPTO-MIT dataset. † indicates that the results are copied from the corresponding papers. "Parallel" indicates whether the model can perform parallel decoding. "Data Augmentation" indicates whether the model is trained by the augmented dataset. The best results are in bold font.

| Category | Model | Top-1 | Top-2 | Top-3 | Top-5 | Top-10 | Parallel | Data Augmentation |
|---|---|---|---|---|---|---|---|---|
| Template-based | Symbolic † | 90.4 | 93.2 | 94.1 | 95.0 | - | ✓ | × |
| Two-stage | WLDN † | 79.6 | - | 87.7 | 89.2 | - | ✓ | × |
| Autoregressive | GTPN † | 83.2 | - | 86.0 | 86.5 | - | × | × |
| | MT-base † | 88.8 | 92.6 | 93.7 | 94.4 | 94.9 | × | × |
| | MEGAN † | 89.3 | 92.7 | 94.4 | 95.6 | 96.7 | × | × |
| | Chemformer † | 91.3 | - | - | 93.7 | 94.0 | × | × |
| | Sub-reaction † | 91.0 | - | 94.5 | 95.7 | - | × | × |
| | Graph2Smiles † | 90.3 | - | 94.0 | 94.8 | 95.3 | × | × |
| | MT † | 90.4 | 93.7 | **94.6** | 95.3 | - | × | ✓ |
| | AT ×100† | 90.6 | **94.4** | - | **96.1** | - | × | ✓ |
| Non-autoregressive | ReactionSink [1] | 91.3 | 93.3 | 94.0 | 94.5 | 94.9 | ✓ | × |
| | NERF | 90.7 | 92.3 | 93.3 | 93.7 | 94.0 | ✓ | × |
| | Our method + NERF | **91.5** | **93.6** | **94.4** | **95.1** | **95.6** | ✓ | × |

## 4.3 OVERALL PERFORMANCE

The overall performance measured by Top-K accuracy is reported in Table 1. We can see that our method improves the NERF model, resulting in the highest Top-1 to Top-10 accuracy. The top-1 accuracy of our method is 91.5%, which is higher than all other non-autoregressive and autoregressive models. This indicates that removing prior in our method can help avoid introducing noise latent variable compared to the original NERF model, thus enforcing the model to fit the dataset better. The Top-2 to Top-10 accuracy of our method is consistently higher than all other non-autoregressive models and competitive with autoregressive models. When considering top-2 and top-3 accuracy, our method is higher than all autoregressive models without data augmentation. This demonstrates that our method can help generate diverse products while achieving precise Top-1 prediction. It is worth to mention that the computational efficiency of our method is nearly the same as the NERF model and better than another non-autoregressive method ReactionSink.

## 4.4 ABLATION STUDIES

To further investigate the effect of each component of our method, we conduct ablation studies. The results are shown in Table 2. We can observe that removing prior can have better Top-1 accuracy than with it but lose diversity. To introduce diversity, we apply boosting training and dropout strategies and the performance of these two strategies is better than the original NERF model. The performance of the combination of these three strategies is the best. This demonstrates that each strategy is beneficial to the model and they can compensate each other and help achieve better performance.

Table 2: The impact of Removing prior, Boosting, Dropout, and the combination of these three strategies for the USPTO-MIT dataset. The unsolvable case is indicated by -.

| Model Name | Top-1 | Top-2 | Top-3 | Top-5 | Top-10 |
|---|---|---|---|---|---|
| NERF | 90.7 | 92.3 | 93.3 | 93.7 | 94.0 |
| Removing prior of NERF | **91.5** | - | - | - | - |
| Removing prior + Boosting | **91.5** | 93.4 | 93.9 | 94.2 | 94.2 |
| Removing prior + Dropout | **91.5** | 93.0 | 93.7 | 94.1 | 94.2 |
| Removing prior + Boosting + Dropout | **91.5** | **93.6** | **94.4** | **95.1** | **95.6** |

## 4.5 THE DISTRIBUTION OF PREDICTED PRODUCTS

We also investigate whether our method can help generate a more precise set of plausible products compared to NERF. For the test set, we group the reactions by the number of predicted products and compute the percentage and HitRate (how many ground-truth products are identified in the predicted products) for each group. Figure 4(a) shows that our method predicts more accurate plausible products compared to NERF, aligning more closely with the real-world scenario where

---

[1]Our method can be also applied to ReactionSink. Due to no open source code and the instability of ReactionSink, we haven't tested our method with it. We will test this once the code of ReactionSink is released.

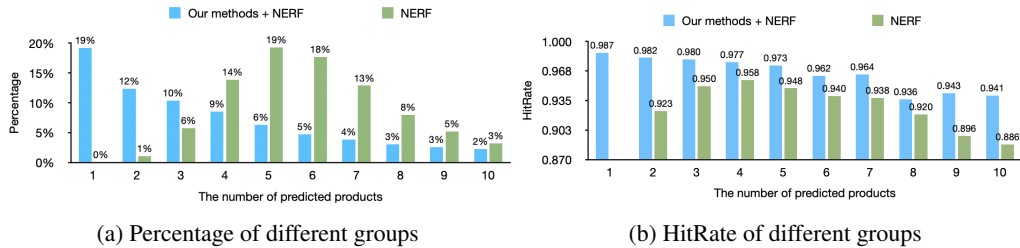

(a) Percentage of different groups

(b) HitRate of different groups

Figure 4: The Percentage and Hitrate of test reactions grouped by the number of predicted products.

the quantity of plausible products adheres to a power-law distribution. The HitRate of our method for each group (in Figure 4(b)) is higher than that of the NERF model, indicating our method can generate more precise products.

### 4.6 THE UTILIZATION DISTRIBUTION OF BOOSTING MODELS

Since we have many model variants produced by boosting training, we evaluate the utilization of each boosting model by calculating the percentage of test samples each infers. As shown in Figure 5, we observe that except for the last model-88, the utilization percentage of most models is around 2%, indicating the boosting models are load balanced. The percentage of model-88 is high mainly because at the end of boosting training, all remaining training reactions are associated with this model which is more easily to be selected by similarity gating.

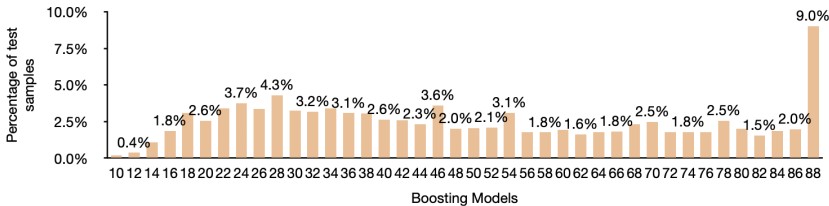

Figure 5: Utilization distribution of boosting models.

### 4.7 EVALUATION ON MULTI-SELECTIVITY REACTIONS

We finally evaluate our method on test reactants with multi-selectivity reactions. To simplify, we select the reactants that have more than two products in the USPTO-MIT dataset and compute the reactant-wise HitRate. Note that a higher HitRate would indicate that a higher percentage of the actual (ground truth) products are being correctly predicted. As shown in Table 3, we can see that our method has a higher HitRate than the NERF model, confirming that our method has enhanced capability for predicting multi-selectivity reactions.

Table 3: The HitRate of our method and NERF model in multi-selectivity reactions.

| Model Name | HitRate@2 | HitRate@3 | HitRate@5 | HitRate@10 |
|---|---|---|---|---|
| NERF | 0.305 | 0.312 | 0.315 | 0.315 |
| Our method + NERF | **0.315** | **0.338** | **0.348** | **0.358** |

## 5 CONCLUSION

Reaction Prediction task is a fundamental and challenging task. In this paper, we first identify the non-uniform uncertainty problem in the reaction prediction which is ignored by all previous non-autoregressive methods. To tackle this challenge, we replace the common generation CVAE architecture with a well-designed generation framework via Boosting and Dropout. Boosting training can obtain large-difference models to capture the large-range uncertainty while dropout can obtain small-different models to capture the small-range uncertainty. We also designed a simple ranking method to rank the predicted products of boosting and dropout to ensure the predicted products are as precise as possible. Experiment results show our model can consistently outperform the baselines.

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
