# OpenReview forum: "Modeling non-uniform uncertainty in Reaction Prediction via Boosting and Dropout"
_ICLR.cc/2024/Conference — ICLR 2024 Conference Withdrawn Submission_

### Official Review · Reviewer_VFvZ · 2023-10-30

**Soundness:** 3 good
**Presentation:** 4 excellent
**Contribution:** 2 fair
**Rating:** 5
**Confidence:** 5

**Summary:**

This paper explores the important uncertainty modeling problem in non-autoregressive reaction prediction. The overall architecture mainly combines the variant Ada-boosting approach with the dropout technique to generate and rank diverse predicted products. Experimental results demonstrate the effectiveness of the proposed approach.

**Strengths:**

(1) The core idea of the proposed approach is clearly written. Both figures clearly illustrate the major claims of this work. The paper structure is also clearly organized;

(2) The proposed approach is a feasible approach to inject uncertainty into the non-autoregressive modeling. Meanwhile, this reviewer can speculate tremendous engineering efforts behind this work;

(3) The addressed challenge in this paper is significant for the non-autoregressive reaction prediction. If the top-k accuracies of the non-autoregressive models can be comparable with autoregressive ones, then there will be an important paradigm shift in reaction prediction research.

**Weaknesses:**

(1) The proposed method is very engineering-heavy and hyper-parameter-tuning-heavy. There are four modules that rely on hyper-parameter selections. The first hyper-parameter is the threshold that determines whether the reaction prediction is correct. The second hyper-parameter is the number of iterations to determine whether a sample should be filtered. The third hyper-parameter is the association mapping between samples and associated models. Last but not least, the warm-up operation trains the model "to some extent" to ensure the basic performance of models. In real application scenarios, it would be very hard to tell to what extent the model reaches the subtle balance between accuracy and diversity;

(2) Based on the current method description, it seems additional (total#iteration/update_step) models must be stored for the following inference works. If it is so, then this reviewer is deeply concerned about the scalability of the proposed method when dealing with much larger reaction datasets in the future. Although a single model in the proposed method might be more lightweight than NERF due to the removal of latent variables, the additional storing space complexity is definitely non-negligible;

(3) Currently it is still unclear how each model is associated with each sample. The words in the paper are "accurately predict". But in what criteria to evaluate the "accurately predict"? What if there are many models that can accurately predict a certain reaction sample? Considering there is also a warm-up process, it is not surprising that most of the models can accurately predict most of the training samples. Then how to determine which model to choose? This reviewer does not find any clear description of this process and this reviewer is deeply concerned about this process since the association criteria might be kind of ambiguous;

**Questions:**

(1) If the iteration is 100, and the update iteration is 2, does it mean the proposed method needs to store 50 models? If so, how could the proposed method scale to larger datasets with much more iterations?

(2) For those hyper-parameter selection sections reminded above, how do you determine the set of hyper-parameters? In what criteria? At this time, the hyper-parameter selections lack informative instructions. This reviewer believes that the final results could be very sensitive to different hyper-parameters.

(3) During inference, it seems an additional reaction-matching process is required. Each testing sample needs to compare itself with 400K reaction samples to determine the top-N similar reactions, which is extremely expensive to compute. If doing parallel decoding, this reviewer can expect a testing sample vector matrix of 40k*d multiplied by a training sample vector matrix of 400K*d, which is nearly unacceptable for larger systems. So is there any approximation approach to do some trade-off between accuracy and speed in terms of inference?

(4) For the product ranking, this work adopts the order "the major model's predictions", "the related model's predictions", "the major model with dropout's predictions", and "the related models with dropout's predictions". It seems this ranking is only based on intuition instead of any quantification approach. Could the authors quantify the uncertainty of different model predictions? Otherwise, in real applications, is it possible that the third "the major model with dropout's predictions" should be ranked higher than "the related model's predictions"? (Just for example). In general, is there any theoretical guarantee to rank the model predictions in this way?

---

### Official Review · Reviewer_ywGc · 2023-11-01

**Soundness:** 3 good
**Presentation:** 3 good
**Contribution:** 3 good
**Rating:** 5
**Confidence:** 3

**Summary:**

This work proposes to tackle reaction prediction (in the forward direction) by training a deterministic predictive model combined with boosting and dropout to handle uncertainty in the reaction outcome. The authors outline a comprehensive framework to tailor boosting to the particular task at hand, and perform experiments on the USPTO-MIT dataset to showcase the effectiveness of their method.

**Strengths:**

(S1): The paper addresses an important problem of reaction prediction, which is a crucial stepping-stone towards multi-step retrosynthetic planning (to verify the steps taken in the backward direction). The proposed approach is reasonable and practical, without being overly complicated for novelty's sake.

(S2): The results on USPTO-MIT are promising, although how impactful these really are is not yet clear to me – see (W1) below.

(S3): The paper is relatively clear and includes informative figures e.g. Figure 2.

**Weaknesses:**

(W1): The results in Table 1 are promising, but would benefit from more analysis. The authors highlight the comparison with other non-autoregressive models, noting how they outperform other models in this class. If being non-autoregressive in itself were beneficial, then one could imagine a practitioner may restrict their model choice to this class, and then the presented method would be SotA; however, I believe non-autoregressiveness does not directly yield any benefits, but rather it can mean the model is faster than e.g. an autoregressive Transformer. Indeed, existing Transformer-based models such as Chemformer [2] are impractically slow [1]. To highlight the proposed method being more efficient, Table 1 could also report inference speed (perhaps when running with batch size of 1), which is important if the model were to be used to assess the feasibility of steps taken during retrosynthetic search [1]. Reporting inference time may strengthen this work, as it may turn out that the proposed method lies on a speed-accuracy Pareto front for all top-k, despite not having the best top-k values overall.

(W2): The discussion of the field and the related work is rather skewed.

- (a) In the abstract, the authors claim that the VAE-based approach is "typically employed" for reaction prediction. In contrast, I would argue the most common approach to forward reaction prediction are still end-to-end language models such as Chemformer [2], closely followed by template-based approaches such as LocalTransform [3]; non-autoregressive electron redistribution methods such as NERF are newer and (in my view at least) slightly less established among practitioners. There is nothing wrong with developing the NERF thread of modelling further, but I think the discussion should be adjusted to reflect the state of the field more accurately.

- (b) The "Reaction Prediction" paragraph in the "Related Work" section may confuse some readers, as it doesn't make a distinction between forward and backward reaction prediction. Models based on end-to-end languge models like [2] can usually work in both directions, but many other models cannot, as one can make stricter assumptions in the forward direction that unlock more interesting model classes. I would suggest clarifying that the authors are concerned with forward prediction (the authors may also want to double-check that the references are indeed all about forward prediction).



=== Nitpicks ===

Below I list nitpicks (e.g. typos, grammar errors), which did not have a significant impact on my review score, but it would be good to fix those to improve the paper further.

- "(2) In high uncertainty scenarios" -> "in" should not be capitalized here

- Missing space before parenthesis in the caption of Figure 2f

- "generalization of the neural networks", "1 denotes the single bond and 2 denotes the double bond" -> I would drop "the" in all of these

- "Follow the preprocessing of the NERF method" -> "Following"?

- The presentation of results in Table1 where the best result within each model class is bolded does not highlight the overall SotA. Maybe it would be better to bold the overall SotA for each top-k and underline the best result within each model class.

- "We implement our model by Pytorch" -> I'd say "using" or "with" instead of "by"

- "combines the graph encoder and transformer encoder" -> I think the second "encoder" should be "decoder"

- In Section 4.3 it would be better to consistently stick to either "top-k" or "Top-k" (capitalized or not); perhaps the non-capitalized version is more common

- "small-different models" - do you mean "small-difference"?



=== References ===

[1] Maziarz et al, "Re-evaluating Retrosynthesis Algorithms with Syntheseus"

[2] Irwin et al, "Chemformer: A Pre-Trained Transformer for Computational Chemistry"

[3] Chem et al, "A generalized-template-based graph neural network for accurate organic reactivity prediction"

**Questions:**

(Q1): Could you elaborate on the values being summed in Equation 1? Is $w_{ij}^{+d}$ the probability that the electron change is _exactly d_ or _at least d_?

(Q2): What happens with $i < t$ in Equation 4?

(Q3): Could you clarify "ranking of product candidates is determined based on the positional order of the prediction models $f_i$"?

(Q4): The paper mentions that the proposed method is as fast as NERF. How is this possible given that many models are boosted and also many forward passes with different dropout masks are utilized? It would be useful to discuss this further, and also report exact inference times (see (W1)).

(Q5): Is the number of products reported in Figure 4a the number of distinct answers that the model produces? If so it may be useful to clarify, as one could alternatively understand this as the average number of product molecules within a single answer (this wouldn't make much sense here as I assume data preprocessing may have converted all reactions to single-product ones, but nevertheless this alternative interpretation could lead to confusion).

---

### Official Review · Reviewer_KPLt · 2023-11-01

**Soundness:** 1 poor
**Presentation:** 2 fair
**Contribution:** 2 fair
**Rating:** 3
**Confidence:** 3

**Summary:**

The work identifies the non-uniform uncertainty problem in the conditional VAE (CVAE) framework of the non-autoregressive reaction prediction (NERF) method, and proposes to improve this framework by better modeling the non-uniform uncertainty, via:
* removing the prior sampling process from the CVAE architecture;
* developing a training method inspired by Adaboost to build a collection of models $F=\\{f_1, f_2,\\dots,f_n\\}$, each $f_i$ is trained on a different subset of samples — this design choice is intended to capture the multimodality in each conditional distribution of product given reactant;
* applying MC Dropout to capture smaller-range uncertainties around each mode of products;
* designing a ranking module to route reactants to the appropriate models in $F$ and to obtain the top product candidates during inference.

The work provides experiment results on the current largest public reaction prediction dataset USPTO-MIT, showing the effectiveness of their method over four different categories of baseline models (template-based, two-stage, autoregressive, and non-autoregressive), via the metrics of Top-K accuracy and HitRate.

**Strengths:**

1. It’s interesting to take a multi-level view over uncertainties, and provide different treatments on the large-range uncertainty and the small-range uncertainty.
2. The design of experiments is clear: it not only demonstrates the efficacy of the proposed method over the baseline models, but also shows each design choice (as the modification over NERF) gradually builds up the improved results.

**Weaknesses:**

1. The description of the main limitation of CVAE is inaccurate: the work mentions that CVAE “cannot manage non-uniform uncertainties effectively” due to the application of “an independent prior (unrelated to the reactants)” — this is not true, since the application of using a simple noise distribution and mapping it through a neural network to generate samples with multimodality is universal across all types of generative models. One can even tell via Figure 2 (c), in which CVAE has been able to model both uni-modal (second reactant from left) and multi-modal (first reactant from the left) conditional distributions. In other words, CVAE can conceptually model the non-uniform uncertainty, and poor performance illustrated in Figure 2 (c) might just be due to the mis-training of the model. As a result, the motivation for discarding the prior from CVAE is not valid.
2. The main components of the proposed method, in terms of key words, are CVAE, boosting, and dropout. However, there are modifications being made to each component:
* A CVAE without independent prior is just a conditional autoencoder.
* It’s understandable that the training of $F$ is called boosting, because each $f_i$ is trained on essentially different sets of training samples — this part is similar to Adaboost. However, boosting algorithms have several distinct characteristics, including: a) they form an ensemble of weak learners to construct a strong predictor; b) all weak learners are combined to provide a strong point estimate. However, the proposed method in this work doesn’t utilize a weak learner for each $f_i$, and each $f_i$ by itself is sufficient (strong enough) to provide a final prediction (when chosen by the ranking/gating module); when combined together, the prediction list shall be diverse rather than a point estimate. There are enough differences between the proposed method and the key properties of a boosting algorithm, that it might be misleading to term this module as boosting.
* The incorporation of dropout during inference has been proposed by [(Gal and Ghahramani, 2016)](https://arxiv.org/pdf/1506.02142.pdf) as MC Dropout, which is not mentioned in this work.

Therefore, it might be a good practice to name each model component in a more straightforward fashion (conditional autoencoder, and MC Dropout), or to describe the component as it is (for example, an Adaboost-inspired procedure for training a collection of models, etc.)

**Questions:**

1. Could the authors elaborate on how the ordering of $f_i$ during inference is done? It’s mentioned in Section 3.5 that the ordering is by the count of training molecules, but Figure 3 indicates that the count is of models rather than of training samples.
2. Could the authors explain why they would “subtract the predictive top-k accuracy by $0.3\\%$ as the final accuracy of our model” (Section 4.1)?
3. In Figure 4 (a), different numbers of predicted products seem to draw different conclusions for the proposed method vs. NERF. Could the authors make a comparison for number 2 and 5, and interpret the difference such that the proposed method has a higher percentage for 2 but lower for 5?

---

### Author Response · Authors · 2023-11-17
**Thanks to all reviewers and withdraw paper for further improving**

We would like to express our sincere thanks to the anonymous reviewers for their insightful feedback. Your comments are very valuable for us and we decided to withdraw our current version of the submission and further improve the paper.